# The Role of MicroRNAs in Bone Metabolism and Disease

**DOI:** 10.3390/ijms21176081

**Published:** 2020-08-24

**Authors:** Yongguang Gao, Suryaji Patil, Airong Qian

**Affiliations:** 1Laboratory for Bone Metabolism, Xi’an Key Laboratory of Special Medicine and Health Engineering, Key Laboratory for Space Biosciences and Biotechnology, Research Center for Special Medicine and Health Systems Engineering, NPU-UAB Joint Laboratory for Bone Metabolism, School of Life Sciences, Northwestern Polytechnical University, Xi’an 710072, China; gaoyongguang@nwpu.edu.cn (Y.G.); suryajip@mail.nwpu.edu.cn (S.P.); 2Department of Chemistry, Tangshan Normal University, Tangshan 063000, China

**Keywords:** bone metabolism, non-coding RNAs, miRNAs, osteoporosis, therapeutics

## Abstract

Bone metabolism is an intricate process involving various bone cells, signaling pathways, cytokines, hormones, growth factors, etc., and the slightest deviation can result in various bone disorders including osteoporosis, arthropathy, and avascular necrosis of femoral head. Osteoporosis is one of the most prevalent disorders affecting the skeleton, which is characterized by low bone mass and bone mineral density caused by the disruption in the balanced process of bone formation and bone resorption. The current pharmaceutical treatments such as bisphosphonates, selective estrogen receptor modulator, calcitonin, teriparatide, etc., could decrease the risk of fractures but have side-effects that have limited their long term applications. MicroRNAs (miRNAs) are one of many non-coding RNAs. These are single-stranded with a length of 19–25 nucleotides and can influence various cellular processes and play an important role in various diseases. Therefore, in this article, we review the different functions of different miRNA in bone metabolism and osteoporosis to understand their mechanism of action for the development of possible therapeutics.

## 1. Introduction

Bone metabolism is a multifarious process that involves various bone cells such as bone marrow mesenchymal stem cells, osteoblasts, osteoclasts, as well as osteocytes and various signaling pathways, including Wnt/*β*-catenin signaling pathway, tumor growth factor-*β (*TGF-*β*)/Smad2/COL4A1 signaling pathway, bone morphogenetic protein (BMP)/suppressor of mothers against decapentaplegic (Smad) pathway, etc. The slightest deviation in the regulation of these pathways or cells can result in various bone disorders such as osteoarthritis, osteopenia, osteopetrosis, osteoporosis, etc. Osteoporosis has emerged as a major and quite common skeletal disorder that leads to fragile bone in post-menopausal women and elderly people and makes them vulnerable to bone fractures [1,2]. Osteoporosis is characterized by low bone mineral density (BMD) due to dysregulation in the process of osteoblast-mediated bone formation and osteoclast-mediated bone resorption. This process is regulated by genetic as well as non-genetic factors such as estrogen deficit, calcium deficiency, physical activity, aging, diet, smoking, etc. [3]. Osteoporosis is often regarded as a disease of women. Yet, one in four men is at the lifetime risk of having an osteoporotic fracture [4]. Osteoporosis in men, in contrast to postmenopausal osteoporosis, is commonly a secondary condition. Half of the osteoporosis cases presented in men are related to underlying disease, hypogonadism, alcohol abuse, or excess glucocorticoid, and in up to 30–50% of male cases, osteoporosis is idiopathic [5,6].

There are various reports to suggest a variety of approaches to reduce and treat the risks of fracture due to osteoporosis such as the use of non-pharmacological as well as pharmacological interventions. A diet supplemented with suitable calcium and vitamin D, physical activities, and fall prevention, quit smoking, etc., are involved in the non-pharmacological approach [7]. Pharmacological interventions such as bisphosphonates (e.g., alendronate, ibandronate, and risedronate), selective estrogen receptor modulator, calcitonin, raloxifene, and teriparatide in postmenopausal osteoporosis woman were shown to have a significant decrease in risk of fractures and tolerable side-effects [8]. However, these interventions have side-effects such as high cost, gastrointestinal disorders, osteonecrosis of the jaw and ear canal, hypophosphatasia (HPP), symptomatic bone marrow edema (BME), etc. [9,10,11]. 

Therefore, suitable therapeutics with little or no side-effects have become an attractive choice. One such therapeutic that could have promising application is gene therapy, which involves the introduction of DNA or RNA into the cell to treat various disorders such as inherited disorders, cancers, osteoporosis, etc. [12]. For that purpose it is necessary to understand the gene profile of coding as well as non-coding RNAs during the diseases, which are significantly changed during the diseases.

MiRNAs (miRNAs) are one such non-coding RNAs that play an important role in many diseases such as cancer, cardiovascular diseases osteoporosis, etc., and the work of many researchers shows the potential of miRNAs in diagnosis and therapeutics of human diseases and use as biomarkers [13].

MiRNAs are single-stranded non-coding RNAs with 19–25 nucleotides in length, which are copied from intergenic, intronic, or polycistronic loci by RNA polymerase II to pri-miRNAs. These pri-miRNAs are consequently processed first in the nucleus and then in the cytoplasm by the Drosha–DGCR8 complex, a member of the RNase III family to pre-miRNA hairpin, and by the Dicer–TRBP complex to miRNA duplexes, respectively. MiRNA duplexes are then integrated into miRNA-induced silencing complex (miRISC) and then relaxed. The mature strand in miRISC is retained while another strand of the duplex is degraded [14,15] (Figure 1). The action through which miRNAs act involves this mature strand and target mRNA. MiRNAs control gene expression either by inhibiting the translation of messenger RNA (mRNA) or by encouraging the degradation of this mRNA [16].

In general, miRNAs bind to the specialized sequence located at the 3′ UTR or 5′ UTR or within the coding region or promoter regions of their target mRNAs and promote deadenylation and decapping of mRNA. This leads to repression of translation and ultimately inhibition of pathways [17]. This influence makes them an ideal target for drug development without side-effects. Therefore, in this review, we summarized the role of miRNA (miRNA) in bone metabolism bone marrow mesenchymal stem cells (BMSCs), osteoclasts and osteoblasts, and osteoporosis and the potential applications of these miRNAs in osteoporosis therapeutics.

## 2. MiRNAs Involved in BMSC Regulation

Bone marrow mesenchymal stem cells (BMSCs) are multipotent cells with capabilities such as self-renewal and differentiation into various cell types such as osteogenic, adipogenic, chondrogenic, etc., and therefore are a common choice in regenerative medicine [18].

Various cellular processes are regulated by mitogen-activated protein kinases (MAPKs) such as extracellular-signal-regulated kinase (ERK), c-Jun N-terminal kinase (JNK), and p38 [19]. Guo et al. reported that the upregulation of miR-214 could efficiently promote the adipocyte differentiation of BMSCs in vitro and reduce osteogenic differentiation. Furthermore, the use of JNK and p38 inhibitors, SP600125 and SB202190, respectively, were reported to enhance the negative effects of miR-214 overexpression on osteogenic differentiation in the BMSCs [20]. Tumor necrosis factor-*α* (TNF-*α*) is a well-recognized inhibitory factor in BMSC osteogenic differentiation and was shown to induce miR-23b expression, which could lower Runt-related transcription factor 2 (Runx2) and consequently reduce osteogenesis leading to severe osteoporosis in mice [21]. The transcription factor such as high-mobility group AT-hook 2 (HMGA2) is involved in adipogenesis [22]. It was reported that the upregulation of miRNA let-7 could stimulate osteogenic differentiation of mesenchymal stem cells (MSCs) and inhibit adipogenic differentiation in vitro. In vivo upregulation could also induce bone formation by reducing the expression of HMGA2. However, the downregulation of let-7 inhibited osteogenic differentiation and promoted adipogenic differentiation [23].

Distal-less homeobox 5 (DLX5) is a protein, which along with myocyte enhancer factor-2 (*Mef2*) regulates the expression of Runx2 through homeodomain-response elements (HRE) located within the distal promoter of Runx2 and promotes activation and expression of osteoblast-specific Runx2 enhancer [24,25]. In osteo-induced BMSC, overexpression of miR-339 was reported to reduce osteogenic differentiation by silencing the action of DLX5, while downregulation of miR-339 was able to promote the DLX5 expression [26].

Vitamins such as vitamin A (retinol) also influence bone metabolism. A metabolic product of vitamin A, retinoic acid (RA), in association with BMP-2 increases osteoblast differentiation. RA is produced by oxidation of retinaldehyde whose levels are regulated by retinoic acid-inducible dehydrogenase reductase 3 (DHRS3), thus playing a role in osteogenesis [27]. The study reported that during osteogenic differentiation of hBMSCs, miR-223 was downregulated while the retinol metabolism pathway was activated. The silencing of miR-233 expression was shown to increase the level of DHRS3 and the activity of alkaline phosphatase (ALP) and extracellular matrix calcification [27]. 

Special AT-rich sequence-binding protein (SATB2) is known to be involved in osteogenesis by increasing the levels of bone matrix proteins as well as osteogenic transcription factors such as osterix (Osx) and runt-related transcription factor 2 (Runx2) [28]. The expression of SATB2 is related to several miRNAs such as miR-31, miR-449-5p, etc. According to the study, overexpression of miR-31 considerably reduced the expression of the osteogenic transcription factors in BMSCs through repression of Satb2 protein [29]. It was shown that miR-31 could also regulate the expression of osterix through binding to its 3’ untranslated region, controlling osteogenic differentiation of mesenchymal stem cells (MSC) into osteoblasts [30]. Li et al. described that miR-449b-5 could reduce BMSC differentiation by targeting SATB2 and downregulation of SATB2 eliminated the positive effect produced on osteoblasts differentiation by miR-449b-5p inhibitor [31]. Likewise, the upregulated miR-384-5p in BMSCs was also shown to promote senescence in BMSCs and subsequently inhibit osteogenic differentiation. Moreover, miR-384-5p could inhibit the expression of *Gli2* and in vivo inhibition of miR-384-5p could stop bone loss and improve the osteogenic capacity in aged rats [32]. As fractures are subsequent results of osteoporosis, miRNA such as miR-374b was reported in such cases. The overexpression of miR-374b in MSCs could increase the genes related to osteogenesis through phosphatase and tensin homolog (PTEN). Furthermore, co-overexpression of PTEN and miR-374b was reported to inverse this effect but just partly. Accordingly, miR-374b targeted PTEN to promote fracture healing [33]. 

The energy required during proliferation and differentiation is generally fulfilled by glucose and fatty acids, but glutamine also serves as an energy source. Glutamine is subsequently converted into *α*-ketoglutarate through glutamate, which is essential for bone formation. It is well-known that as the time of osteoblasts differentiation of MSCs increases, the uptake of *L*-glutamine also increases. *L*-glutaminase (GLS) can convert this glutamine into glutamate, and, therefore, the levels of GLS must be maintained. However, it was demonstrated that miR-200a-3p, which was highly expressed, could target GLS and decrease osteogenic differentiation of MSCs [34]. The osteo-induced human bone marrow mesenchymal stem cells (BMSCs) at days 7 and 14 had under-expressed miR-206 and markedly increased glutamine uptake. The overexpression of miR-206 showed weakened expressions of alkaline phosphatase (ALP), osteocalcin (OCN), Runx2, and osteopontin (OPN). Moreover, the targeting of GLS mRNA by miR-206 reduced the expression and metabolism of GLS and glutamine in BMSCs [35].

Thus, by targeting different molecules, miRNAs play a significant role in regulating different aspects of bone marrow mesenchymal stem cells (Table 1).

Many signaling pathways are involved in various bone metabolic processes to maintain tissue homeostasis. Wnt signaling, TGF-*β*/BMP, PI3K/AKT signaling, etc., plays an important role in increasing bone formation and decreasing bone resorption, thus balancing the bone homeostasis [37]. MiRNAs also affect these pathways by targeting different mRNAs and proteins and influence bone remodeling.

### 2.1. MiRNAs Regulating BMSCs through Wnt Signaling Pathway

The increased differentiation of BMSCs into adipogenic cells can lower bone mass and increase the risk of bone fractures. Various miRNAs influence the commitment or favor one of the differentiations of BMSCs by influencing different pathways. The Wnt signaling pathway is a well-known signaling cascade that either depends on the *β*-catenin (Wnt/*β*-catenin pathway) or function independent of it (noncanonical pathways). The receptors such as low-density lipoprotein receptor-related protein (LRP) 5 and 6, and inhibitors like dickkopf-1 (DKK1), sclerostin, and secreted frizzled-related proteins (SFRPs) determine the fate of Wnt signaling by regulating *β*-catenin either for proteasomal degradation or for the nuclear translocation to regulate gene expression. This induces osteoblast differentiation through osterix [38,39]. The function of these receptors and inhibitors can be regulated by different miRNAs (Table 2) targeting different proteins involved in Wnt signaling (Figure 2).

It was reported that during adipocytic differentiation of BMSCs, the expression of miR-204-5p was upregulated, along with the adipogenic transcription factors. This upregulation of miR-204-5p could promote adipogenesis while its knockdown has shown increased osteogenesis of BMSCs. This regulation of adipogenesis by miR-204-5p was due to the control of dishevelled segment polarity protein 3 (DVL3) expression. DVL is known to transmit the Wnt signal, which activates the signaling mechanism that eventually stabilizes the *β-*catenin. Thus, by targeting DVL3, miR-204-5p prevented the activation of the Wnt/-catenin signaling pathway [40,56]. Wnt/*β*-catenin signaling was shown to stimulate osteoblast (OB) differentiation by enhancing Runx2 expression. The study revealed that overexpression of miR-214 could target *β*-catenin to suppress its expression and subsequently the action of the Wnt/*β*-catenin signaling pathway resulting in reduced differentiation of BMSCs [41]. Conversely, the upregulation of miR-101 was reported to promote Wnt gene expression, initiating the Wnt/*β*-catenin signaling pathway and targeting of enhancer of zeste 2 (EZH2), which was shown to inhibit osteogenesis in BMSCs by repressing Wnt and BMP. Thus, it could promote osteogenic differentiation [42,57].

Glycogen synthase kinase-3*β* (GSK-3*β*) is an important mediator in various signaling pathways. The phosphorylation facilitated by GSK3 was shown to activate the destabilization of *β*-catenin. Therefore, GSK3 inhibitors could be prospective therapeutics for human diseases [58]. MiR-346 upregulation was reported to target glycogen synthase kinase-3*β* (GSK-3*β*), activating the Wnt/*β*-catenin pathway and promoting osteogenic differentiation of hBMSCs [43]. MiR-376c is a well-reported tumor suppressor miRNA that inhibits cell proliferation as well as invasion in osteosarcoma. MiR-376c overexpression was shown to have an inhibitory effect on osteoblast differentiation, while with low expression a promotive effect. Furthermore, the over-activation of miR-376c was established to target the Wnt-3 and ADP ribosylation factor guanine nucleotide exchange factor 1 (ARF-GEF-1) causing a decline in canonical Wnt/*β*-catenin signaling and inhibition of osteoblast differentiation [44]. The recent work on the regulators of human mesenchymal stromal cells (hMSCs) has revealed the involvement of miR-33 family members in regulating different pathways and processes of hMSCs. Over-expression of miR-33a-5p was reported to activate SNAIL and SLUG, which regulate YAP/TAZ transcription and post-transcription to control EMT signaling as well as gaining of the osteoblast phenotype in hMSCs, whereas miR-33a-3p inhibition has also shown the same results. Furthermore, miR-33a-3p targeted YAP to regulate osteoblast phenotype in the Nh-Ost cell through epidermal growth factor receptors (EGFR) signaling [45].

The canonical Wnt signaling antagonists are also targeted by miRNAs such as dickkopf-1 (DKK1) was shown to be targeted by overexpressed miR-433-3p to promote osteoblast differentiation [46]. MiR-203 overexpression could also enhance osteogenic gene expression by reducing DKK1 expression and showed the opposite effect on adipogenic gene expression [47,48]; low-density lipoprotein (LDL)-receptor-related protein 5 (LRP5) is one of the co-receptors that is involved in the Wnt pathway. The overexpression of miR-23a was shown to target LRP5, which reduced Wnt signaling and prevented osteogenic differentiation of hBMSCs in vitro [49]. The number of days of hMSC differentiation also affects the expression of miRNAs, for example, miR-1286. MiR-1286 overexpression was reported to lower the expression of Runx2 and OCN as well as ALP activity. The levels of FZD4, an important gene in the Wnt signaling pathway, were positively correlated with the osteogenic marker levels but not with the level of miR-1286. Therefore, the inhibition of hMSC differentiation to osteogenic cells by miR-1286 can be reversed by up-regulating FZD4 [50]. According to the study, the positive correlation between miR-144 and secreted frizzled-related protein 1 (SFRP1) was demonstrated. SFRP1 is an inhibitor of *β*-catenin. When upregulated, miR-144 was reported to target SFRP1 and promote not only proliferation but also differentiation of BMSCs to osteoblasts and inhibit apoptosis by stimulating the Wnt/*β*-catenin pathway [51]. 

The ability of stem cells to differentiate into different cells is governed by different transcription factors. In vitro, SRY (sex-determining region Y)-box 2 (SOX2) and Krüppel-like factor 4 (KLF4) along with other transcription factors have reported to induce pluripotency in the host cell [59]. SOX2 was also identified as an inhibitor of Wnt signaling. Adil Akkouch et al. revealed that the overexpression of miR-200c could lower the expression of SOX2 and KLF4. This lowered expression of SOX2, therefore, could promote Wnt signaling and osteogenic differentiation in human bone marrow mesenchymal stromal cells (hBMSCs), as well as bone regeneration [52]. It was reported that the upregulation of miR-200c could also target myeloid differentiation primary response 88 (Myd88) and thus activate AKT/*β*-catenin signaling to promote osteogenesis in hBMSCs [53]. 

Another study has described the positive role of miR-542-3p in stimulating bone formation by inhibiting the expression of secreted frizzled-related protein-1 (SFRP1) mRNA and protein in osteoporotic rats [54]. In hBMSCs from osteoporosis patients, the inverse correlation of miR-16-2* expression with osteogenic genes was demonstrated. The upregulated expression of miR-16-2* could impair the osteogenic differentiation by binding to Wnt5A and consequently obstruct the Wnt signaling pathway, whereas downregulation rescued this effect possibly due to stimulated activation of Runx2 [55].

### 2.2. MiRNAs Involved in BMSCs Regulation through the TGF-β/BMP Pathway

Transforming growth factor-beta (TGF-*β*)/bone morphogenetic protein (BMP) signaling has various roles in bone formation. Canonical Smad-dependent pathways involving TGF-*β*/BMP ligands, TGF-*β*/BMP receptors, and Smads, and non-canonical Smad-independent signaling pathways such as p38-MAPK are the two main pathways through which the signal is conveyed to regulate the osteoblast differentiation of mesenchymal cell and bone formation [60] (Figure 3).

The overexpression of miR-155 in MSCs was reported to decrease the expression of Runx2, OPN, phosphorylated Smad1/5/8 (p-Smad1/5/8), and OCN and to reduce osteogenesis in vitro as well as in vivo by targeting Runx2 and bone morphogenetic protein receptor 9 [61]. Similarly, miR-765 overexpression in hMSCs also inhibited osteogenic-gene expression as well as phosphorylation of Smad1/5/9 through BMP6 and subsequently osteogenic differentiation of hMSC [62]. Bone morphogenetic protein type Ib receptor BMPR1b is a serine/threonine kinase receptor that plays a vital role in MSC osteogenic differentiation. MiR-125b was demonstrated to target BMPR1b, and in vivo inhibition of miR-125b was shown to repair bone defects [63]. Homeobox a10 (HOXA10) is an abdominal B (Abd B) class of homeobox protein-encoding genes that is induced by BMP2. In response, Hoxa10 stimulates the expression of transcription factor Runx2 that is crucial for bone formation [64]. The upregulation of miR-320a was reported to target HOXA10 and reduce the osteogenic differentiation of hMSCs [65]. The study conducted by Jiang Qiu et al. reported that the expression of transforming growth factor (TGF)-*β*, phosphorylated (p)-Smad2 was decreased while the adipogenic differentiation of BMSCs was enhanced when miR-214-5p was overexpressed [66]. The study further revealed that the TGF-*β* inhibitor and miR-214-5p antagomir also decreased the expression of Smad2, TGF-*β*, and COL4A1 proteins and also suggested that this diminished BMSC osteogenic differentiation might have been through COL4A1 [66].

Biglycan (Bgn) is a class I small leucine-rich proteoglycan (SLRP) that is expressed during cell proliferation and mineralization. By regulating TGF-*β*, Bgn plays an important role in survival as well as in the growth of BMSCs [67]. It was shown that knockout (KO) of miR-185 could increase the osteogenesis in primary osteoblasts collected from miR-185-KO mice as well as improved bone volume in KO mice. Furthermore, it was concluded that miR-185 could target biglycan to stimulate bone formation via the BMP/Smad pathway [68]. Another study reported that when levels of miR-23a-5p were dropped, it promoted the osteogenic differentiation of hBMSCs, while overexpression could repress the differentiation via MAPK13 [69]. MiR-217 was also downregulated during the osteogenic differentiation of BMSCs. Overexpression of miR-217 reduced osteogenesis of BMSCs and lowered the expression of Runx2 by targeting the binding site in the 3′-untranslated region and had an adverse effect on the osteogenic differentiation by altering ERK and p38 MAPK phosphorylation [70]. By targeting Smad7, a negative regulator of Runx2, overexpression of miR-590-5p protected Runx2 protein. This protection stabilized Runx2 and promoted osteoblast differentiation of mouse MSCs [71].

Tob 1 (transducer of ErbB2 1) is a protein known to inhibit BMP/Smad signaling and deficiency, of Tob can increase the bone mass. [72] According to the work of Yan Li et al. in OVX-induced osteoporosis, the level of miR-26a was significantly reduced. The overexpression of miR-26a was able to rescue the in vitro osteogenic differentiation of MSCs isolated from OVX mice as well as ectopic bone formation in vivo by targeting Tob1 and thus promoting BMP/Smad signaling [73].

Transforming growth factor-beta receptor 1 (TGFBR1) is a serine/threonine kinase receptor that transduces TGF-*β* signaling for the cell surface to the nucleus; it was shown to be targeted by overexpressed let-7a-5p and could reduce the osteogenic differentiation of BMSC as well as ALP activity and the formation of calcified nodules [74]. Glucocorticoids (GCs) are prescribed as immunosuppressive agents for inflammatory diseases or to those who have received organs during organ transplantation. Nevertheless, GC-induced osteoporosis (GIOP) is a result of excess or long term use of GCs; in such cases, studies have reported the low expression of let-7f-5p. Upregulation of let-7f-5p was shown to induce the osteogenic differentiation of dexamethasone (Dex)-induced BMSCs but to lower the expression of TGFBR1. An in vivo study has also shown similar results when let-7f-5p was overexpressed, thus reducing bone loss [75].

PPARγ (peroxisome proliferator-activated receptor-gamma) is another important transcription factor whose expression is an indication of increased adipogenesis. MiR-130a and miR-27b levels were shown to be elevated during human MSCs osteogenic differentiation with Runx2 and osterix, but the levels of adipogenic marker genes such as PPARγ and C/EBP*β* were reported to be lowered [76]. Further study has revealed that the upregulation of miR-130a and miR-27b could negatively regulate the expression of SMAD specific E3 ubiquitin protein ligase 2 (Smurf 2) and could target PPARγ increasing osteogenesis and reducing adipogenesis in human MSC [77].

### 2.3. MiRNAs Involved in BMSCs Fate through Epigenetic Route

Epigenetics play a vital role in regulating various cellular processes, and impairment could lead to severe diseases. MiRNAs are also involved in the regulation of the fate of BMSCs through the epigenetic route. It is well-known that histone deacetylase 4 (HDAC4) deacetylates and degrades Runx2, thereby inhibiting chondrocytes and osteoblast differentiation. The upregulation of miR-29a was reported to increase the osteogenesis of HMSCs by targeting HDAC4 [78]. It was reported that miR-19a-3p expression in osteoporosis patients was decreased and miR-19a-3p overexpression was demonstrated to target HDAC4 and increase the expression of ALP, OCN, and Runx2. This increased expression resulted in stimulated osteogenic differentiation of hMSCs [79]. RICTOR is known to repress the activity of PPARγ, a transcriptional factor involved in adipogenesis, and prevent MSCs adipogenic differentiation. Overexpression of miR-188 targeted histone deacetylase 9 (HDAC9) and RPTOR-independent companion of MTOR complex 2 (RICTOR) and promoted adipogenesis and reduced osteogenesis of BMSCs [80].

### 2.4. MiRNAs Regulating BMSCs Fate through Extracellular Vesicles (EVs)

Extracellular vesicles (EVs) are discharged by most of the bone cells and include not only exosomes and microvesicles but also apoptotic bodies. They carry and transport proteins, growth factors as well as miRNAs to regulate bone formation [81]. Therefore, change in these EVs or their miRNAs relates to age-associated dysregulation in stem cell function. EVs isolated from the interstitial fluid of bone marrow of young (3–4 months) and aged (24–28 months) mice have reported high augmentation of miRNAs; however, the miRNA profile in these EVs differed significantly. Especially, the miR-183 cluster (miR-96/-182/-183) was shown to be highly prevalent in aged EVs compared to young mice. The overexpression of miR-183-5p in BMSCs could ease the cell proliferation and osteogenic differentiation as well as promote senescence and reduce protein levels of heme oxygenase-1 (Hmox1), which was reported as a target of miR-183-5p [82]. Another miRNA, miR-128-3p, was reported in the exosome of mesenchymal stem cells of aged rats. MiR-128-3p overexpression was described to target Smad5 and lower its expression, thus increasing osteogenesis and healing of the bone fracture [83].

The following table summarizes the role of miRNAs in regulating BMSCs through different mechanisms (Table 3).

## 3. MiRNAs Involved in Osteoclast Regulation

The cytoskeleton organization of osteoclast and migrations are important processes that define the homeostasis of bone and progress of osteoporosis. This can be regulated by RNAs. However, the different RNAs are regulated by different miRNAs (Table 4). Calcr (calcitonin receptor) has known to be involved in the regulation of survival and resorption of osteoclast and could be targeted by miR-29 [84]. Furthermore, it was shown that miR-29 also targeted macrophage lineage associated RNAs. Based on knockdown experiments, miR-29 could positively regulate the formation of osteoclasts by targeting RNAs significant not only for the cytoskeletal organization but also for commitment and osteoclast function [84]. Under RANKL stimulation, osteoclast was reported to have increased expression of miR-31. In such osteoclasts, inhibition of miR-31 suppressed osteoclasts formation and bone resorption, as well as the formation of actin rings, and increased RhoA expression which was the target of miR-31. Thus, by regulating RhoA expression, miR-31 controlled the cytoskeleton organization in osteoclasts [85]. Multiple myeloma (MM) cells are implicated in regulating the balance between bone formation and bone resorption and also in stimulating osteoclasts. The upregulation of miR-29b into osteoclasts was demonstrated to impair the activities of not only cathepsin K but also metalloproteinase 9 (MMP9) and formation of actin rings. Additionally, C-FOS and metalloproteinase 2, canonical targets as well as nuclear factor-activated T cells c1 (NAFTc-1), and osteoclast transcription factor levels were also decreased, which led to reduced cell differentiation and function of human osteoclasts [86]. One of the two isoforms of miR-133, miR-133a, was reported in circulating monocytes in humans to promote the differentiation of osteoclasts. In vitro experiments demonstrated that miR-133a expression was upregulated during osteoclastogenesis and stimulated the differentiation of RAW264.7 and THP-1 cells into osteoclasts via receptor activator of nuclear factor-kappa-*β* ligand (RANKL), while in vivo experiments in ovariectomized (OVX) rats have revealed that miR-133a promoted bone loss as well as reduced BMD of the lumbar spine [3,87]. One study revealed the role of miR-218 and miR-618 in the differentiation of RAW264.7 cells into osteoclasts. The up-regulated expression of miR-218 or miR-618 reduced the progression of RAW264.7 cells into osteoclasts in vitro, whereas down-regulation showed the reverse effect. This was attributed to their effect on the TLR-4/MyD88/NF-κB signaling pathway [88]. MiRNAs present in exosomes of BMSCs were also reported to regulate osteoclasts such as miR-31a-5p detected in the exosomes derived from BMSCs of aged rats was shown to enhance osteoclastogenesis, and subsequently bone resorption and inhibition of miR-31a-5p reduced osteoclast differentiation and function and increased RhoA activity [89].

During the osteoclastogenesis of bone marrow-derived macrophages (BMMs), the expression of miR-214 was reported to be increased, and upregulation could promote osteoclast formation. MiR-214 was found to target the PTEN, which is known to regulate osteoclast differentiation from RAW 264.7 when induced by RANKL via the PI3K/Akt pathway. In vivo experiments have also supported these results by lowering PTEN levels, increasing osteoclast activity and inhibiting BMD [90].

## 4. MiRNAs Involved in Osteoblast Regulation

The osteoblasts cells are involved in bone formation and thus maintain homeostasis and metabolism. These cells also play an important role in the development of osteoporosis. Wnt/*β*-catenin and TGF*β* signaling pathways also play an important role in the differentiation of osteoblasts. According to the study, miR-140-3p was shown to respond to the over-expression of the Wnt3a. TGF*β*3, which was increased in response to overexpression of Wnt3a, could be targeted by miR-140-3p. However, when overexpressed, miR-140-3p stimulated the transcription of OCN in MC3T3-E1 cells. Thus, miR-140-3p regulated osteoblast differentiation by acting as a regulatory factor between Wnt3a and TGF*β*3 signaling pathways [92]. According to the study, miR-133a-5p overexpression was shown to significantly lower the expression of osteoblast differentiation markers as well as ECM mineralization in MC3T3-E1 cells by targeting 3′ UTR of Runx2 and reduce Runx2 expression at both mRNA and protein levels [93]. The expression of miR-92a-1-5p was decreased during MC3T3-E1 osteogenic differentiation, and overexpression could target *β*-catenin, a positive regulator of osteogenic differentiation, thereby inhibiting osteogenesis [94]. Similarly, miR-542-3p overexpression could also inhibit osteoblast differentiation, and inhibition could promote the expression of osteoblast-specific genes, ALP activity, and matrix mineralization. MiR-542-3p overexpression suppressed bone morphogenetic protein 7 (BMP-7) and inhibited BMP-7/PI3K-survivin signaling, which reduced osteogenic differentiation and increased osteoblast apoptosis. In vivo miR-542-3p silencing also increased bone formation, bone strength, as well as improved trabecular microarchitecture in OVX mice [95].

During trauma or bone fractures, the translocated lipopolysaccharide (LPS) triggers the inflammatory response and was established to be the main inhibitor of BMP2-induced osteogenic differentiation. In MC3T3-E1 cells, LPS significantly reduced the expression of miR-23b, whereas it upregulated Smad 3. Overexpression of miR-23b was demonstrated to target Smad 3, which had a negative effect on osteogenesis of MC3T3-E1 during LPS treatment [96]. The differentiated osteoblasts can have a high expression of miR-15b. Despite having more than one target genes such as Crim1, Smurf1 (SMAD specific E3 ubiquitin-protein ligase 1), Smad7, and HDAC4 (histone deacetylase 4), Smurf1 expression was increased by miR-15b inhibitor, while protein levels of Runx2 were lowered without affecting mRNA levels. However, during miR-15b mimic treatment, Runx2 protein expression was improved, while that of Smurf1 protein was lowered. Furthermore, Smurf1 was shown to target Runx2 and degrade it by the proteasomal pathway, which would support why Runx2 was reduced during miR-15b inhibition [97].

There are many inhibitors of osteogenic differentiation such as TGF*β*3, HDAC4, ACTVR2A, and CTNNBIP1, an inhibitor of *β*-catenin, and DUSP2, which deactivates ERK and anchors it within the nucleus, etc., which disrupts the signal transduction. Overexpression of miR-29b was able to suppress these inhibitors and promote osteoblast differentiation of the preosteoblast cell line MC3T3-E1 [98]. MiR-375-3p was reported to inhibit osteogenesis by promoting the expression of a well-known negative regulator of osteogenesis, sclerostin (SOST). In osteoblast precursor, MC3T3-E1 cells, upregulation of miR-373-3p could target LRP5 and *β*-catenin, leading to increased SOST and reduced osteogenesis [99].

Therefore, elevated as well as reduced levels of miRNAs in osteoblasts can affect the normal function of cells and in turn bone formation. The following table summarizes the role and different targets of different miRNAs in osteoblasts (Table 5).

### 4.1. MiRNAs Involved in the Osteoblast Regulation under Hypoxia

Hypoxia was demonstrated to promote bone loss as well as inhibit osteogenic differentiation by downregulating Runx2. MC3T3-E1 cells cultured under hypoxic conditions showed elevated levels of miR-21-5p which when overexpressed could downregulate the expression of SMAD7 at the mRNA and protein level, indicating SMAD7 as a target of miR-21-5p, and could also upregulate the protein expression of Runx2 [100], while miR-135-5p upregulation promoted the osteoblast differentiation in MC3T3-E1 cells by targeting hypoxia-inducible factor 1 *α* inhibitor (HIF1AN), which was reported to inhibit differentiation and calcification [101].

### 4.2. MiRNAs Involved in the Osteoblast Regulation under High Glucose Conditions

Patients with diabetes mellitus (DM) also tend to show the increased occurrence of osteoporosis. Moreover, high glucose conditions were shown to reduce TGF-*β* signaling in bone tissue and bone mass and fragility in diabetic osteoporosis (DO). MiR-590-5p and osteoblastic proteins have reported being downregulated when MC3T3-E1 cells were subjected to high glucose (HG) conditions while Smad7 was upregulated. Moreover, miR-590-5p upregulation targeted Smad7 and considerably stimulated HG-treated MC3T3-E1 cell proliferation and differentiation through the upregulation of TGF-*β* signaling [102] (Table 6).

### 4.3. MiRNAs Involved in the Osteoblast Regulation under Microgravity

Bone metabolism is also affected by microgravity influencing different miRNA (Table 7) and can lead to the progression of osteoporosis. Hind-limb unloading and random positioning machine (RPM) are the most commonly employed techniques to simulate microgravity. During simulated microgravity, the level of miR-139-3p was reported to be upregulated, affecting osteoblast differentiation as well as apoptosis, and downregulation of miR-139-3p could enhance the differentiation and reduce apoptosis of MC3T3-E1 cells. ELK1 (ETS like-1 protein) is necessary for miR-139-3p mediated regulation of osteoblast differentiation and apoptosis and was targeted by miR-139-3p [103]; another miRNA whose levels were increased in microgravity was miR-132-3p. It was reported that upregulated miR-132-3p inhibited osteoblast differentiation in part by lowering Ep300 protein expression. E1A binding protein p300 (Ep300) is a type of histone acetyltransferase that is essential for the activity and stability of Runx2, which, consequently, resulted in the repression of the activity and acetylation of Runx2, leading to decreased osteogenesis [104].

Mechanical stretching as well as cyclic stretching and shear stress were reported to modulate the expression of a number of miRNAs that play an important role in various biological and pathological processes. A study has shown that when osteoblasts were subjected to cyclic mechanical stretching (CMS), miR-103a levels were upregulated. This increased expression of miR-103a significantly reduced the Runx2 protein level in osteoblasts. Similarly reduced osteogenesis was observed in hind-limb unloaded mice [105]. The elevated expression of miR-208a-3p in mice under hind-limb unloading (HLU) was negatively associated with bone formation in HLU mice. In vitro miR-208a-3p upregulation could reduce osteoblast differentiation, whereas silencing reversed the effect. MiR-208a-3p targeted activin A receptor type I (ACVR1) to adversely regulate differentiation of osteoblasts. However, in vivo pre-treatment with antagomiR-208a-3p not only increased bone formation and trabecular microarchitecture but also partly recovered bone loss resulted due to mechanical unloading [106]. There are many miRNAs such as miR-138-5p that were reported to respond to the different mechanical stimuli and also regulate the different aspects of osteogenesis. It was reported that miR-138-5p was inversely related to bone formation and targeted microtubule actin crosslinking factor 1 (MACF1), which was established to be required for the proliferation and differentiation of osteoblasts. Moreover, the pre-treatment of antagomir-138-5p partially rescued the bone loss in HLU mice caused due to the unloading [107,108,109].

## 5. MiRNAs Involved in Osteoporosis

In osteoporosis patients, the serum provides an ideal sample for the detection of various miRNAs, which might be upregulated or downregulated, and can be used as biomarkers (Table 8). It has established that miR-365a-3p was overexpressed in serum of osteoporosis patients but the levels were decreased gradually as osteo-induction prolonged in hBMSCs [110]. The downregulation of miR-365a-3p was shown to reduce the OCN, OPN, and collagen I expression; the overexpression can reduce the potential of hBMSCs for mineralization. Further investigation has revealed that the miRNA-365a-3p targeted Runx2 and consequently stimulated osteoporosis progression [110]. According to another study, highly expressed miR-579-3p in osteoporosis patients regulated Sirtuin 1 (Sirt1) expression. Sirt1 is very well expressed in MSCs, which affects the multipotency of MSCs. By reducing the Sirt1 level, miR-579-3p subsequently inhibited the differentiation of hMSCs into osteogenic cells. However, combined overexpression of both miRNA-579-3p and Sirt1 could rescue this effect [111]. In the serum of osteoporosis patients as well as in bone marrow mesenchymal stem cells (BMSCs) isolated from aged humans and mice, miR-96 was suggested to be up-regulated. This increased expression of miR-96 decreased osteogenic differentiation of BMSCs in vitro as well as in vivo, while inhibition of miR-96 mitigated the bone loss caused by age by targeting osterix miR-96 [112].

The alkaline phosphatase (ALP) is a homologous dimer protein, and each monomer consists of 449 amino acids. It is an indicator of enhanced osteogenic activity. The activation and expression of ALP are regulated by the activation of transcription factor 3 (ATF3) by binding to its promoter. The overexpression of ATF3 can lead to lowered ALP activity and consequently lower osteogenic activity. The low expressed miR-27a-3p, when overexpressed, was shown to target ATF3. Thus, miR-27a-3p promotes osteogenic differentiation of hMSCs providing miR-27a-3p as a positive factor in promoting hMSCs differentiation into osteoblasts [113]. Growth factors such as fibroblast growth factors (FGFs) and FGF receptors (FGFRs) are also important for the development of bone. For tissue repair, cranial bone intramembranous ossification, and in cranial suture these factors play an important role [116]. Any abnormal signaling can lead to various skeletal anomalies. The study reported that the MSCs isolated from osteoporotic mice showed up-regulated expression of miR-214, which could inhibit osteoblast differentiation in vitro by inhibiting the FGFR1/FGF pathway, while downregulation of miR-214 could promote osteogenic differentiation [36]. SLC39A1 encodes ZIP1, a zinc transporter essential for zinc transport into the cell, which is vital for proliferation and differentiation of osteoblast-like cells. According to a study conducted by Lv et al., miR-133 expression was increased in estrogen-deficiency induced osteoporosis and also during MSC osteogenic differentiation [114]. Furthermore, miR-133 could target SLC39A1 and lower its expression in mMSCs and hMSCs isolated from an OVX mouse model and bone marrow of PMOP patients, respectively, thus reducing osteogenesis [114]. Myocyte enhancer factor 2c (Mef2c) transcription factor was suggested to play an important role in maintaining the differentiated state of muscle cells. It was proposed that miR-27a expression was considerably reduced in osteoporosis postmenopausal women and was shown to be essential for MSC differentiation. Further investigation has revealed that silencing of miR-27a could reduce the bone mass in OVX mice by targeting Mef2c [115].

Another study showed that overexpression of miR-214 in BMMS isolated from OVX mice reduced BMMs differentiation into osteoclasts by targeting Rab27a, which is essential in maintaining osteoclast functions and phenotypes [117]. Microphthalmia-associated transcription factor (MITF) is known to be an essential factor in the induction of hematopoietic stem cells into osteoclasts. Overexpression of miR-340 was described to reduce osteoclast differentiation of BMM by inhibiting the expression of MITF [118]. The differentiation of RAW 264.7 cells into osteoclasts was shown to be inhibited upon overexpression of miR-218 through suppression of NF-κB signaling by targeting tumor necrosis factor receptor 1 (TNFR1), which is required for TNF to initiate nuclear factor-κB (NF-κB) [119]. Similarly, miR-145 overexpression was also reported to suppress osteoclast activity in OVX mice through SMAD3 expression [120] (Table 9).

The serum of postmenopausal osteoporosis patients was reported to often show a high concentration of miR-320a. This high expression could influence cell apoptosis in pre-osteoblast MC3T3-E1 cells suppressing the viability and differentiation of MC3T3-E1 cells by targeting microtubule-associated protein 9 (MAP9) and controlling the PI3K/AKT signaling pathway [121]. In the serum of osteoporotic postmenopausal women, miR-181c-5p and miR-497-5p levels were downregulated. The same was also found in the bone tissues of OVX and aging mice. Furthermore, upregulation was reported to increase the osteogenic differentiation as well as mineralization of hFOB1.19 and MC3T3-E1 cells [122]. Axin1 is an intracellular inhibitor of canonical Wnt signaling and a part of the *β*-catenin destruction complex [123]. MiR-539 was described as decreased in osteoblasts as well as osteoclasts of osteoporotic rats, and overexpression could target the axin1. This led to the upregulation of gene expression of various markers in osteoblasts, as well as apoptotic markers in osteoclasts, indicating the positive role of miR-539 in proliferation and differentiation of osteoblasts and the negative role in osteoclastogenesis [91]. The MCF2L (MCF.2 cell line derived transforming sequence like) gene coding guanine nucleotide exchange factor is involved in the Rho/Rac signaling pathways, which could be involved in apoptosis. It was established that increased levels of miR-140-3p in osteoporosis patients could lower the cell viability of MC3T3-E1 cells. The inhibition of overexpressed miR-140-3p was shown to target MCF2L and reduce apoptosis of MC3T3-E1 cells and enhance cell viability and differentiation [124]. The serum of elderly women with fractures and OVX mice was also described to show higher levels of exosomal miR-214-3p derived from osteoclasts. In vitro transfection of miR-214-3p in osteoblast was shown to reduce the levels of osteogenic markers such as ALP, OPN, etc., whereas in vivo knock-in also reduced bone formation but inhibition improved the bone formation in OVX mice [125]. The expression of miR-133a was upregulated in fracture nonunion patients and MC3T3-E1 cells, and miR-133a overexpression could reduce the Runx2 and BMP2 levels and prevent the bone formation and fracture healing [126] (Table 10).

## 6. Conclusions and Perspective

The disruption in the coordinated processes between BMSCs, osteoblasts, or osteoclasts can have a significant effect on bone metabolism. These coordinated processes are governed by many different molecules, including miRNAs. MiRNAs, due to their ability to bind and degrade specific mRNAs, control the expression of genes and thereby affect cell function and play an important role in various bone diseases such as osteoporosis. Recent studies on osteoporosis have revealed that miRNAs circulating in the serum of such patients can be detected in a varying expression and, therefore, can act as biomarkers, and these expressions of miRNAs can have a significant role in the function of BMSCs, osteoblasts, and osteoclasts cells. However, there are some downsides to the miRNA as biomarkers such as lack of standardization in choosing the source of circulating miRNAs, lack of large sample size, lack of comprehensive study of a specific disease, and the effect of age, gender, and incapability to precisely identify the origin of the miRNAs. Many studies use blood as a source for miRNA, but blood cells make it difficult to identify the origin of the specific miRNA [127]. However, TAmiRNA, a European leader in miRNA diagnostics, showed the clinical applications of their licensed miRNAs as biomarkers in osteoporosis, and their lead product OsteomiR™ could provide the risk of the first fracture in postmenopausal osteoporosis and type-2 diabetes women [128].

However, when it comes to the transitioning of miRNA therapeutics into clinical study, it faces two major problems; first is stability, which is often limited due to nuclease attack on naked oligonucleotides [129]. The solution to this problem is chemical modifications, but these also have side effects such as off-targeting and decreased miRNA activity. The second problem is poor cell penetration. The delivery systems provide protection against nucleases and increases the chances of cell delivery, but care has to be taken to reduce the toxicity of the material used toward cells [130].

Many miRNAs involved in bone tissue are also present in different cells, tissues, and even in other diseases, raising the question of specificity in developing miRNA therapeutics. Hence, novel delivery systems that can only target specific tissue/cells are being developed to increase the efficiency and efficacy of miRNA therapeutics using cell-penetrating peptides [131,132,133].

The next part of the miRNA study could be the clinical applications of miRNA. There are many miRNA therapeutics that have successfully entered phase I/II clinical trials in oncology [134]. Moreover, further studies are warranted to determine if miRNA-based therapeutics can be used in pre-clinical and clinical trials for bone-related diseases, providing a basis for safe and large-scale application.

Apart from the miRNAs mentioned in the current article, there are still many more non-coding RNAs that remain to be identified, and their detailed role in the progression of osteoporosis remains to be elucidated. Therefore, further research in eliminating such disadvantages can certainly make miRNAs as potential targets for developing new therapeutics in diagnosing and treating various diseases.

## Figures and Tables

**Figure 1 ijms-21-06081-f001:**
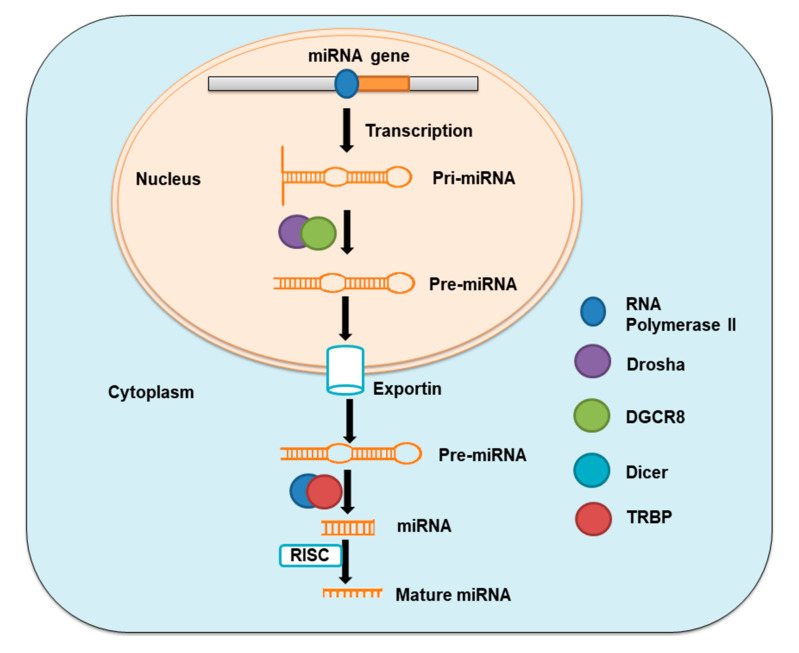
Schematic of miRNA synthesis.

**Figure 2 ijms-21-06081-f002:**
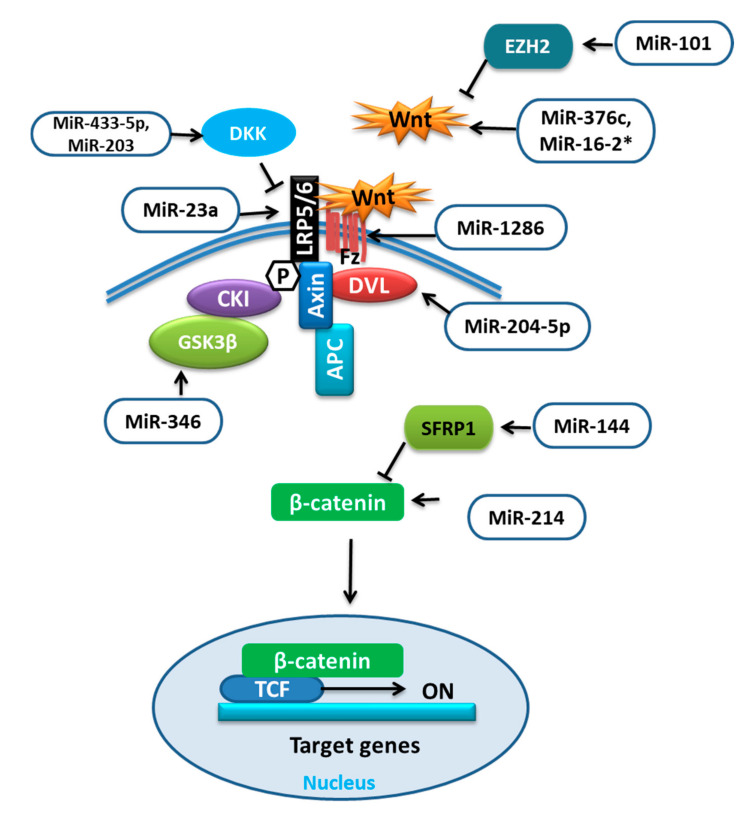
miRNAs targeting different components of Wnt signaling.

**Figure 3 ijms-21-06081-f003:**
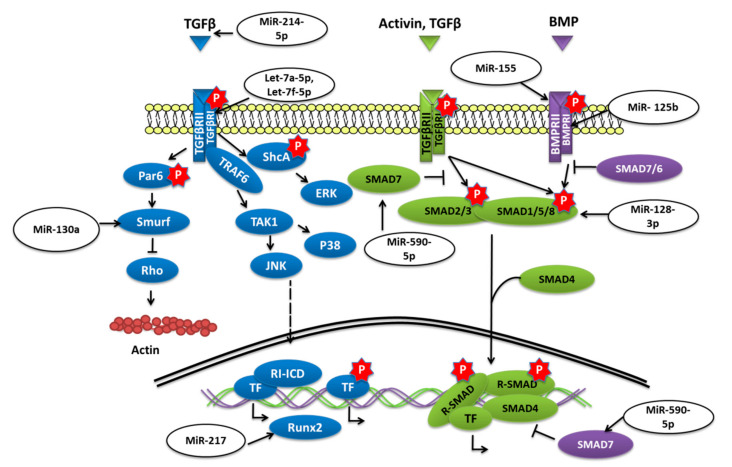
miRNAs targeting different components of TGF-*β*/SMAD signaling.

**Table 1 ijms-21-06081-t001:** Role and target of miRNAs in BMSC regulation.

MiRNA	Relative Expression	Role	Target	References
MiR-214	Upregulation	Inhibits osteoblast differentiation and enhances adipogenesis	*β*-catenin, FGFR1/FGF, JNK, and p38	[20,36]
MiR-23b	Upregulation	Reduces osteogenesis	Runx2	[21]
*Let-7*	Upregulation	Stimulates osteogenic differentiation of MSCs	HMGA2	[23]
MiR-339	Upregulation	Impairs osteogenic differentiation BMSCs	DLX5	[26]
MiR-223	Downregulation	Stimulates osteogenic differentiation of hMSCs	DHRS3	[27]
MiR-31	Upregulation	Reduces osteogenic differentiation of BMSCs	Satb2 protein, osterix	[29,30]
MiR-449b-5p	Upregulation	Reduces BMSCs differentiation	Satb2	[31]
MiR-384-5p	Upregulation	Promotes senescence and decreases osteogenesis of rat BMSCs	Gli2	[32]
MiR-374b	Upregulation	Promotes osteogenic differentiation and fracture healing	PTEN	[33]
MiR-200a-3p	Upregulation	Impairs osteogenic differentiation of MSCs	*L*-glutaminase	[34]
MiR-206	Upregulation	Impairs osteogenic differentiation BMSCs	*L*-glutaminase	[35]

**Table 2 ijms-21-06081-t002:** BMSCs regulation through the Wnt signaling pathway.

MiRNA	Relative Expression	Role	Target	References
MiR-204-5p	Upregulation	Increases adipogenic differentiation of BMSCs	DVL3	[40]
MiR-214	Upregulation	Inhibits osteoblast differentiation	*β*-catenin	[41]
MiR-101	Upregulation	Promotes osteogenic differentiation	EZH2	[42]
MiR-346	Upregulation	Promote osteogenic differentiation of hBMSCs	GSK-3*β*	[43]
MiR-376c	Upregulation	Inhibits osteoblast differentiation	Wnt-3 and ARF-GEF-1	[44]
MiR-33a-5p	Upregulation	Maintains osteoblast phenotype in hMSCs and Nh-Ost cell	SNAIL and SLUG,	[45]
MiR-33a-3p	Downregulation	Maintains osteoblast phenotype in Nh-Ost cell	YAP	
MiR-433-3p	Upregulation	Promotes osteogenic differentiation	DKK1	[46]
MiR-203	Upregulation	Enhance osteogenic differentiation of MSCs	DKK1	[47,48]
MiR-23a	Upregulation	Prevents osteogenic differentiation of hBMSCs	LRP5	[49]
MiR-1286	Upregulation	Reduces differentiation of hMSCs	FZD4	[50]
MiR-144	Upregulation	Promote proliferation and differentiation of bone marrow-derived mesenchymal stem cells to osteoblasts	SFRP1	[51]
MiR-200c	Upregulation	Induces osteogenic differentiation in hBMSCs	Sox2, Klf4, Myd88	[52,53]
MiR-542-3p	Upregulation	Stimulates bone formation	SFRP1	[54]
MiR-16-2 *	Upregulation	Impairs osteogenic differentiation	WNT5A	[55]

* Star/passenger microRNA.

**Table 3 ijms-21-06081-t003:** MiRNAs involved in BMSC regulation.

TGF-*β*/BMP Pathway				
MiRNA	Relative Expression	Role	Target	References
MiR-155	Upregulation	Reduces osteogenesis in vitro as well as in vivo	Runx2, BMPR2	[61]
MiR-765	Upregulation	Inhibits osteogenic differentiation of hMSC	BMP6	[62]
MiR-125b	Downregulation	Promotes repair of bone defects	BMPR1b	[63]
MiR-320a	Upregulation	Inhibits osteogenic differentiation of hMSCs	HOXA10	[65]
MiR-214-5p	Upregulation	Promotes the adipogenic differentiation over the osteogenic differentiation of BMSCs	TGF-*β*	[66]
MiR-185	Downregulation	Stimulates osteogenesis	Biglycan	[68]
MiR-23a-5p	Downregulation	Promoted osteogenic differentiation of hBMSCs	MAPK13	[69]
MiR-217	Upregulation	Impairs osteogenic differentiation	Runx2	[70]
MiR-590-5p	Upregulation	Promotes osteoblast differentiation of mouse MSCs	Smad7	[71]
MiR-26a	Upregulation	Increases bone formation	Tob1	[73]
Let-7a-5p	Upregulation	Decreases osteogenic differentiation of BMSC	TGFBR1	[74]
Let-7f-5p	Upregulation	Induce the osteogenic differentiation of dexamethasone (Dex)-induced BMSCs	TGFBR1	[75]
MiR-130a and miR-27b	Upregulation	Promotes osteogenic differentiation of BMSC	Smurf2 and PPARγ	[76,77]
**Epigenetic route**				
MiR-29a	Upregulation	Promotes osteogenesis	HDAC4	[78]
MiR-19a-3p	Upregulation	Promotes osteogenic differentiation of hMSCs		[79]
MiR-188	Upregulation	Promotes adipogenesis and reduced osteogenesis of BMSCs	HDAC9, RICTOR2	[80]
**Extracellular vesicles (EV)**				
MiR-183-5p	Upregulation	Reduces BMSC cell proliferation and osteogenic differentiation	Hmox1	[82]
MiR-128-3p	Upregulation	Increases osteogenesis and healing of the bone fracture	Smad5	[83]

**Table 4 ijms-21-06081-t004:** MiRNAs involved in osteoclast regulation.

MiRNA	Relative Expression	Role	Target	References
MiR-29	Upregulation	Regulates the formation of osteoclast	Calcr	[84]
MiR-31	Downregulation	Suppresses osteoclast formation and bone resorption	RhoA	[85]
MiR-29b	Upregulation	Lowers cell differentiation and function of osteoclasts	NAFTc-1	[86]
MiR-133a	Upregulation	Promotes osteoclastogenesis and bone loss in vivo	RANKL	[87]
MiR-218 and miR-618	Upregulation	Reduces osteoclastogenesis	TLR-4/MyD88/NF-κB signaling	[88]
MiR-31a-5p	Downregulation	Reduces osteoclasts differentiation and function	RhoA	[89]
MiR-214	Upregulation	Increases osteoclast activity	PTEN	[90]
MiR-539	Upregulation	Reduces osteoclastogenesis	Axin1	[91]

**Table 5 ijms-21-06081-t005:** MiRNAs involved in osteoblast regulation.

MiRNA	Relative Expression	Role	Target	References
MiR-140-3p	Upregulation	Regulates osteoblast differentiation	TGF*β*3	[92]
MiR-133a-5p	Upregulation	Lowers the expression of osteoblast differentiation markers	Runx2	[93]
MiR-92a-1-5p	Upregulation	Negatively regulates osteogenic differentiation of MC3T3-E1	*β*-catenin	[94]
MiR-542-3p	Upregulation	Inhibits osteoblast differentiation	BMP-7	[95]
MiR-23b	Upregulation	Promotes osteogenic differentiation in MC3T3-E1 cells	Smad 3	[96]
MiR-15b	Upregulation	Promotes osteoblast differentiation	Crim1, Smurf1, Smad7, and HDAC4	[97]
MiR-29b	Upregulation	Promotes osteoblast differentiation of MC3T3-E1	TGF*β*3, HDAC4,	[98]
MiR-375-3p	Upregulation	Inhibits osteogenesis in MC3T3-E1	LRP5 and *β*-catenin	[99]

**Table 6 ijms-21-06081-t006:** MiRNAs involved in osteoblast regulation under different conditions.

MiRNA	Relative Expression	Role	Target	References
**Hypoxia**				
MiR-21-5p	Upregulation	Promotes osteoblast differentiation of MC3T3-E1	SMAD7	[100]
MiR-135-5p	Upregulation		HIF1AN	[101]
**High Glucose (HG)**				
MiR-590-5p	Upregulation	Promotes osteoblast differentiation of MC3T3-E1	Smad7	[102]

**Table 7 ijms-21-06081-t007:** MiRNAs involved in osteoblast regulation under microgravity.

MiRNA	Relative Expression	Role	Target	References
MiR-139-3p	Downregulation	Increases differentiation and reduces apoptosis of MC3T3-E1 cells	ELK1	[103]
MiR-132-3p	Upregulation	Inhibits osteoblast differentiation	Ep300	[104]
MiR-103a	Upregulation	Inhibits bone formation	Runx2	[105]
MiR-208a-3p	Upregulation	Reduces osteoblast differentiation	ACVR1	[106]
MiR-138-5p	Upregulation	Inhibits osteoblast differentiation and bone formation	MACF1	[107]

**Table 8 ijms-21-06081-t008:** Role of miRNAs in osteoporosis BMSCs.

MiRNA	Relative Expression	Role	Target	References
MiR-365a-3p	Downregulation	Impairs osteogenic differentiation of hMSCs	Runx2	[110]
MiR-579-3P	Upregulation		Sirt1	[111]
MiR-96	Upregulation	Impairs osteogenic differentiation of BMSCs	Osterix	[112]
MiR-27a-3p	Upregulation	Promotes osteogenic differentiation of hMSCs	ATF3	[113]
MiR-133	Upregulation	Promote osteogenesis of mMSCs and hMSCs	SLC39A1	[114]
MiR-27a	Downregulation	Reduces the bone mass in OVX mice	Mef2c	[115]

**Table 9 ijms-21-06081-t009:** MiRNAs involved in osteoporosis osteoclasts.

MiRNA	Relative Expression	Role	Target	References
MiR-214	Upregulation	Reduces BMM differentiation into osteoclasts	Rab27a	[117]
MiR-340	Upregulation	Reduces osteoclast differentiation factor	MITF	[118]
MiR-218	Upregulation	Inhibits differentiation of RAW 264.7 cells into osteoclasts	TNFR1	[119]
MiR-145	Upregulation	Suppresses osteoclast activity in OVX mice	SMAD3	[120]

**Table 10 ijms-21-06081-t010:** Role of miRNAs in osteoporosis osteoblasts.

MiRNA	Relative Expression	Role	Target	References
MiR-320a	Upregulation	Promotes cell apoptosis in MC3T3-E1	MAP9	[121]
MiR-181c-5p and miR-497-5p	Upregulation	Promotes the differentiation and mineralization of hFOB1.19 and MC3T3-E1	−	[122]
MiR-539	Upregulation	Promotes proliferation and differentiation of osteoblasts	AXIN1	[91]
MiR-140-3p	Upregulation	Suppresses MC3T3-E1 viability and differentiation	MCF2L	[124]
MiR-214-3p	Upregulation	Reduces osteoblast differentiation and bone formation in vivo	−	[125]
MiR-133a	Upregulation	Delays the healing of fractures	Runx2/BMP2	[126]

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
