# Peer review of "The Role of MicroRNAs in Bone Metabolism and Disease"

_ijms, 2020, doi:10.3390/ijms21176081_

Round 1

Reviewer 1 Report

The authors have provided an overview of the role of miRNAs in maintaining bone homeostasis and approaches targeting miRNAs to establish a healthy bone remodeling during diseases such as osteoporosis. The approaches described in the manuscript are interesting and are well-presented. It will provide a good review to researchers working in the field of molecular and skeletal biology in order to examine the potential for miRNA-based new therapies in the treatment of bone disorders.

The authors need to improve the manuscript by including the scope for future diagnostics and therapeutics. The authors need to provide information from the existing data if there are any miRNA(s) reported so far that have/has been always detected consistently as a diagnostic marker(s) under bone diseases (or animal models of bone loss) with independent experiments. There are some discordant reports that also speculate some inconsistencies in the specificity of miRNAs. However, the current manuscript sounds to be written in the light of considering only those reports that strengthen the potential for therapeutic applications of miRNA. Therefore, I do feel that the authors missed several key points that underlie the pitfalls in developing miRNAs as diagnostics or therapeutics. Further, occurrence of similar miRNAs in various tissues, cell types and also in other unrelated diseases could invalidate the use of miRNAs specificity to bone disease. The authors need to discuss the current challenges and multiple anomalies in miRNAs that hinder their development as biomarker(s) and/or therapeutic target(s) to treat bone disorders. 

Author Response

Reviewer’s comment 1- The manuscript entitled “MicroRNAs- Role in bone metabolism and disease” by Yongguang Gao et al is of potential interest for the readers of the International Journal of Molecular Sciences. A review article is provided. The authors aim to summarize current knowledge regarding the role of microRNAs in bone metabolism and disease. The article begins with a short but sufficient introduction in which the complexity of bone metabolism and the nature and formation of microRNAs are briefly described. This is followed by various sections, each of which discusses a particular aspect of bone metabolism with regard to known microRNAs. Each of these sections contains a lot of relevant information, which has also been presented in graphical and tabular form. This is a good indication of the amount of work invested in this manuscript. Random checks of the references did not reveal any problems.

Author’s response 1- We thank you for your valuable time, efforts, and your kind and valuable suggestions.

Reviewer’s comment 2- My main criticism is that the order of the sections is not entirely clear. It starts with MSC, followed by extracellular vesicles, certain signaling cascades, then again cells (osteoblasts), diseases such as osteoporosis etc. My suggestion is either to better explain the sequence of the sections or to re-sort them according to certain aspects. For example, one could first discuss the different cell types (MSC, osteoblasts, osteoclasts), then the signaling pathways (WNT, BMP), and finally the role of microRNAs in different pathologies such as osteoporosis. Apart from this, a very comprehensive and valuable work is provided.

Author’s response 2- We apologizes for not presenting the content articulately. Now the content has been categorized into 6 sections. Sections 2, 3, and 4 have become miRNAs involved into BMSCs, osteoclasts, and osteoblasts, respectively with subsections.

We have designed and made the section to avoid redundancy. If we had discussed the different cell types such as, MSC, osteoblasts, osteoclasts, then the signaling pathways (WNT, BMP), again we would had to write the microRNAs involved in signaling pathways that have been studied in the different cells which would have raised the question why they were not included under the section of MSC/osteoblasts/osteoclasts. Therefore, we decided to include the miRNAs that are involved in MSC/osteoblasts/osteoclasts regulation through different pathways, conditions, etc. under the main section of MSC/osteoblasts/osteoclasts.

However, we have taken it into consideration your latter suggestion about osteoporosis and have prepared the different section as ‘5. MiRNAs involved in osteoporosis’.

Reviewer 2 Report

The manuscript entitled “MicroRNAs- Role in bone metabolism and disease” by Yongguang Gao et al is of potential interest for the readers of the International Journal of Molecular Sciences.

A review article is provided.   The authors aim to summarize current knowledge regarding the role of microRNAs in bone metabolism and disease. The article begins with a short but sufficient introduction in which the complexity of bone metabolism and the nature and formation of microRNAs are briefly described. This is followed by various sections, each of which discusses a particular aspect of bone metabolism with regard to known microRNAs.

Each of these sections contains a lot of relevant information, which has also been presented in graphical and tabular form. This is a good indication of the amount of work invested in this manuscript. Random checks of the references did not reveal any problems. My main criticism is that the order of the sections is not entirely clear. It starts with MSC, followed by extracellular vesicles, certain signalling cascades, then again cells (osteoblasts), diseases such as osteoporosis etc.

My suggestion is either to better explain the sequence of the sections or to re-sort them according to certain aspects. For example, one could first discuss the different cell types (MSC, osteoblasts, osteoclasts), then the signalling pathways (WNT, BMP), and finally the role of microRNAs in different pathologies such as osteoporosis.

Apart from this, a very comprehensive and valuable work is provided.

Author Response

The authors MicroRNAs- Role in bone metabolism and disease” reviewed an important and new topic for mineral metabolism.

We thank you for providing your valuable time and efforts to help us improve this manuscript and also for your kind comments

Reviewer’s comment 1- In the text to make it easier for the reader to understand the mechanisms regulated by miRNAs, it could be useful to make subchapters describing the works that report miRNAs in mice compared to those in humans

Author’s response 1- Thank you for your kind suggestions. But we apologize for the fact that we decided not to make the subsections because of the two reasons.

  1. There are many miRNAs that have been shown to present in mice as well as humans.
  2. Moreover, some microRNAs share the same target in mice and humans.

Therefore, we thought it would be prudent to mention them alongside to make it easy to understand and compare rather than making the separate subsections.

Reviewer’s comment 2- Are there any paper that simultaneously evaluated the circulating miRNAs with those contained in the microvesicles to see if there are any differences?

Author’s response 2- To the best of our knowledge and through web search we could find the articles that specifically studied the comparison of circulating miRNAs and those contained in the microvesicles. However, we agree that this idea of comparative study could have the significant potential to the contribution in miRNA therapeutics.

Reviewer’s comment 3- Why are lines 314-135 in bold, as are lines 443-444 and 470?

Author’s response 3- We thank you for bringing it to our attention and apologize for the mistake. We have made corrections to the mistakes.

Reviewer 3 Report

The authos "MicroRNAs- Role in bone metabolism and disease" reviewed an important and new topic for mineral metabolism. Some tips could improve the paper:

  1. In the text to make it easier for the reader to understand the mechanisms regulated by mi RNAs, it could be useful to make subchapters describing the works that report miRNAs in mice compared to those in humans
  2. Are there any paper that simultaneously evaluated the circulating miRNAs with those contained in the microvesicles to see if there are any differences?
  3. Why are lines 314-135 in bold, as are lines 443-444 and 470?

Author Response

Reviewer’s comment 1-The authors have provided an overview of the role of miRNAs in maintaining bone homeostasis and approaches targeting miRNAs to establish a healthy bone remodeling during diseases such as osteoporosis. The approaches described in the manuscript are interesting and are well- presented. It will provide a good review to researchers working in the field of molecular and skeletal biology in order to examine the potential for miRNA-based new therapies in the treatment of bone disorders.

Author’s response 1- We thank you for providing your valuable time and efforts to help us improve this manuscript and also for your kind comments.

Reviewer’s comment 2- The authors need to improve the manuscript by including the scope for future diagnostics and therapeutics.

Author’s response 2- Thank you for your kind suggestion. We have taken it into consideration and have included the scope of miRNA diagnostics and therapeutics which can be seen from line numbers 644-648 in concluding remarks.

Reviewer’s comment 3- The authors need to provide information from the existing data if there are any miRNA(s) reported so far that have/has been always detected consistently as a diagnostic marker(s) under bone diseases (or animal models of bone loss) with independent experiments. There are some discordant reports that also speculate some inconsistencies in the specificity of miRNAs.

Author’s response 3- To the best of our knowledge and through web search we could find the articles that mention the consistent presence of miRNA as a biomarker. However, we did find the report stating the clinical applications of licensed miRNAs as biomarkers in osteoporosis by But TAmiRNA, which can be seen at line number 629-632.

Reviewer’s comment 4- However, the current manuscript sounds to be written in the light of considering only those reports that strengthen the potential for therapeutic applications of miRNA. Therefore, I do feel that the authors missed several key points that underlie the pitfalls in developing miRNAs as diagnostics or therapeutics. Further, occurrence of similar miRNAs in various tissues, cell types and also in other unrelated diseases could invalidate the use of miRNAs specificity to bone disease.

Author’s response 4- We have explained the disadvantages of miRNA therapeutics in our concluding remarks. It is true that the presence of similar miRNAs in different tissues and cell and even in other diseases can obstruct the development of possible miRNA therapeutics. These parts have been explained in the conclusion and perspectives section which can be seen from line numbers 639-643.

Reviewer’s comment 5- The authors need to discuss the current challenges and multiple anomalies in miRNAs that hinder their development as biomarker(s) and/or therapeutic target(s) to treat bone disorders.

Author’s response 5- We thank you for bringing it our attention. We have included the current challenges that hinder the development of miRNA as biomarker(s) and/or therapeutic target(s) to treat bone disorders in our concluding remarks from line numbers 625-629, 633-638.

Round 2

Reviewer 1 Report

Accept. The manuscript in its revised form is acceptable for publication.